# Joint Cross-Consistency Learning and Multi-Feature Fusion for Person Re-Identification

**DOI:** 10.3390/s22239387

**Published:** 2022-12-01

**Authors:** Danping Ren, Tingting He, Huisheng Dong

**Affiliations:** 1School of Information & Electrical Engineering, Hebei University of Engineering, Handan 056038, China; 2Hebei Key Laboratory of Security & Protection Information Sensing and Processing, Handan 056038, China

**Keywords:** person re-identification, deep learning, feature classifier, cross-learning, attention mechanism

## Abstract

To solve the problem of inadequate feature extraction by the model due to factors such as occlusion and illumination in person re-identification tasks, this paper proposed a model with a joint cross-consistency learning and multi-feature fusion person re-identification. The attention mechanism and the mixed pooling module were first embedded in the residual network so that the model adaptively focuses on the more valid information in the person images. Secondly, the dataset was randomly divided into two categories according to the camera perspective, and a feature classifier was trained for the two types of datasets respectively. Then, two classifiers with specific knowledge were used to guide the model to extract features unrelated to the camera perspective for the two types of datasets so that the obtained image features were endowed with domain invariance by the model, and the differences in the perspective, attitude, background, and other related information of different images were alleviated. Then, the multi-level features were fused through the feature pyramid to concern the more critical information of the image. Finally, a combination of Cosine Softmax loss, triplet loss, and cluster center loss was proposed to train the model to address the differences of multiple losses in the optimization space. The first accuracy of the proposed model reached 95.9% and 89.7% on the datasets Market-1501 and DukeMTMC-reID, respectively. The results indicated that the proposed model has good feature extraction capability.

## 1. Introduction

Person re-identification research [1,2] has undergone a paradigm shift from traditional methods to deep learning methods. The traditional person re-identification research contains two parts: image feature extraction and distance metric. Among them, feature extraction is used to obtain more discriminative and recognizable person image information in the image and person re-identification methods based on feature representation, primarily by obtaining color information of person images, LBP (Local Binary Pattern) [3], SIFT (Scale Invariant Feature Transform) [4], and other features with discriminative power of features. The aim of similarity metrics is to use appropriate metric functions to reduce the intra-class distance of person features while increasing the inter-class distance of person features with different identities, such as LMNN (Large Margin Nearest Neighbor classification) [5,6], XQDA (Cross-view Quadratic Discriminant Analysis) [7], and so on. However, traditional person re-identification research methods have limited ability to extract image features.

In recent years, image feature extraction methods have made significant breakthroughs based on deep learning. Liu et al. [8] constructed a topological learning network framework augmented by integrated contextual information to enhance the ability to extract multi-scale features from the model and to mine valid information complementary to pedestrian appearance features to improve feature representation. Zheng et al. [9] proposed an IDE (ID-discriminative Embedding) network with a residual network as the backbone and the person’s ID as the training label, which has become an important baseline in the field of person re-identification. Hermans et al. [10] used a triplet loss function to train the model, and it is common for existing deep learning methods to jointly use ID loss and triplet loss to optimize the model. Sun et al. [11] proposed a local feature-based person re-identification method (Part-based Conv Baseline) PCB, in which the output features are sliced in the horizontal direction and the chunked features are stitched together when calculating the similarity distance. Zheng et al. [12] proposed a multi-granularity pyramid model combining different granularity features, but the image chunking method still caused alignment problems between local features. Luo et al. [13] proposed the AlignedReid model to dynamically adjust local features to solve the problem of unaligned pedestrian pose that exists in real scenes; however, it consumes more resources when computing the features. Zheng et al. [14] used a pose estimation model to predict the key points of the human body, but this approach necessitates the addition of another pose estimation model, which reduces the model’s generalization performance. The SENet attention mechanism network model proposed by Hu et al. [15] compresses each channel into a weight on the feature map after dimensionality reduction and finds the relationship with the feature map from the channel dimension. Fu et al. [16] proposed to fuse local features, global features, and multi-granularity features to improve person re-identification accuracy; however, they neglected the impacts of occlusion, pose, and other factors. To avoid the attention mechanism from being overly focused on the foreground, Chen [17] combined both channel and spatial information, but the regions extracted by combining individual attention networks lacked adequate semantic interpretation and could not cover all discriminative image feature information.

To address the problems of the abovementioned methods, this paper investigated the extraction of pedestrian image features. We designed a model joint cross-consistency learning and multi-feature fusion for the person re-identification model CCLM (Cross Consistent Learning and Multi-feature) for person re-identification. The main work of this paper is as follows.

(1)The ResNet50 residual network [18] was used as the backbone network, and the interference of invalid feature information was reduced by embedding the dual-attention mechanism module Convolutional Block Attention Module (CBAM) [19] and the mixed-pooling module (MPM) in the residual network.(2)In the cross-consistency learning module of the model, the model was pre-trained to design image classifiers specific to different viewpoints to extract features from different viewpoints and reduce the impact of viewpoint transformation on the extraction of image information.(3)The model was fine-tuned in the multi-feature fusion module, and the fused multi-level features were used to match the similarity with images from the image library to enhance the feature representation capability of the model.(4)In the process of model optimization, for the difference of multiple losses in the optimization space, Cosine Softmax loss [20] was introduced to eliminate the spatial inconsistency of cross-entropy loss and triplet loss, and Cluster Center Loss (CCL) was proposed to make the model focus on intra-class distance as well as inter-class distance in the optimization process.

This paper proposed the CCLM, a joint cross-consistency learning and multi-feature fusion person re-identification model, to address the drawbacks of the earlier approaches.

## 2. Algorithm Flow

In this section, the joint cross-consistency learning and multi-feature fusion for person re-identification network model are presented in five parts, starting with the general structure of the network, followed by the attention mechanism used by the network, followed by the mixed pooling module and then the cross-consistency learning module of the network, and finally explaining the feature pyramid module of the network. The modules collaborate with each other to improve the ability of the model to extract features.

### 2.1. Network Structure

In order to extract the image detail information CCLM person re-identification model is based on the ResNet50 residual network, which contains an attention mechanism module, a mixed-pooling module MPM, CCL, FPN, and a multi-feature fusion module. The network model proposed in this study is shown in Figure 1.

The attention mechanism module is located in layers 2 through 5 of the backbone network, while the down-sampling operation in the last layer is removed, as well as the global average pooling and fully connected layers in the original residual network; after layer 5, it is proposed to use mixed pooling rather than average pooling to improve the network’s ability to extract image features.

The training procedure was divided into two steps: cross-consistency learning and multi-feature fusion. The cross-consistency learning module randomly divides the person re-identification dataset into two groups based on the camera view and then designs a classifier to train the network model for each of the two groups of images with different camera views. The model was then pre-trained by cross-using classifiers with different specific information to eliminate the effect of different camera view shifts on the model’s extraction of valid information.

After pre-training the model with certain knowledge, the multi-feature fusion module takes the feature vectors obtained from the pyramid module and the hybrid pooling module as input, and finally optimizes the backbone network, the pyramid module, and the multi-feature fusion module by combining the Cosine Softmax loss, the triplet loss, and the cluster center loss.

### 2.2. Attention Mechanism Module

The CBAM module consists of two sub-attention modules: the channel attention module and the spatial attention module. The structure of the channel attention module is depicted in Figure 2, where the features extracted by the backbone network are weighted appropriately by the channel attention module and the spatial attention module in turn.

The feature map *F*(*x*) of the image *x* extracted in the backbone network is input to the max pooling and average pooling layers to obtain two feature vectors $F_{Max}\left(x\right)$ and $F_{Avg}\left(x\right)$ respectively, and then the two feature vectors are input to the multilayer perceptron with shared parameters; finally, the corresponding channel attention mapping matrix $M_{c}\left(F\left(x\right)\right)\in R^{C\times 1\times 1}$ is generated after the Sigmoid activation operation. The one-dimensional attention mask for the channel domain is calculated as Equation (1):(1)$$\begin{array}{l} M_{c}\left(F\left(x\right)\right)=sigmoid\left(MLP\left(Max\left(F\left(x\right)\right)\right)+MLP\left(Avg\left(F\left(x\right)\right)\right)\right)\\ \;\;=sigmoid\left(W_{1}\left(W_{0}\left(F_{Max}\left(x\right)\right)\right)+W_{1}\left(W_{0}\left(F_{Avg}\left(x\right)\right)\right)\right) \end{array}$$
where *F*(*x*) denotes the image features of the backbone network input, *Max*(***)and *Avg*(*) denote the maximum and average pooling, *MLP*(*)denotes multilayer perception operation, *W*_0_ ∈ R^C/r×C^ and *W*_1_ ∈ R^C×C/r^. To obtain, weight the mapping matrix obtained after the channel attention module with the corresponding elements of the original input feature map in the channel dimension.
(2)$$F^{\prime }\left(x\right)=Mc\left(F\left(x\right)\right)⊗F\left(x\right)$$
where *F′*(*x*) is the feature map obtained after channel attention.

The spatial attention model is similar to channel attention, as shown in Figure 3. The procedure for calculating the two-dimensional attention mask for the spatial domain is Equation (3):(3)$$\begin{array}{l} M_{s}=sigmoid\left(conv\left(\left[Max\left(F^{\prime }\left(x\right)\right);Avg\left(F^{\prime }\left(x\right)\right)\right]\right)\right)\\ \;\;=sigmoid\left(conv\left(\left[{F^{\prime }}_{Max}\left(x\right);{F^{\prime }}_{Avg}\left(x\right)\right]\right)\right) \end{array}$$
where ${F^{\prime }}_{Max}\left(x\right)$ ∈ *R*^1×H×W^, ${F^{\prime }}_{Avg}\left(x\right)$ ∈ *R*^1×H×W^, and *Conv* represents the convolution operation. Following that, in the spatial dimension, the spatial attention feature mapping matrix and the input feature vector *F′*(*x*) are element-wise multiplied to obtain the final image feature map representation as Equation (4):(4)$$F^{\prime \prime }\left(x\right)=Mc\left(F^{\prime }\left(x\right)\right)⊗F^{\prime }\left(x\right)$$

Finally, the attention module’s weighted image feature map *F″*(*x*) is used as input to the deeper network.

### 2.3. Mixed Pooling Module

To extract global personal features while also highlighting differences between the person and the background, in this paper, we proposed mixed pooling instead of average pooling in residual networks, which compensates for the shortcomings of average pooling in residual networks and improves the ability of the person re-identification model to extract detailed features from images by preserving the whole body information of the person and deepening the person outline while paying more attention to the differences between persons and the background.

As shown in Figure 4, the vector *P_Avg_* obtained from the average pooling of the image features is subtracted from the vector *P_max_* obtained from the maximum pooling, and the subtracted features are computed using a 1 × 1 convolution kernel, after which the difference features between *P_Avg_
*and *P_max_
*are obtained by passing through the Batch Normalization(*BN)* and *Relu* layers respectively. The *P_max_
*features obtained by maximum pooling are summed with the different features between *P_Avg_
*and *P_max_
*using the same computational process to obtain the 2048-dimensional feature vector *P_mix_*.

The $P_{mix}$ is calculated as Equation (5):(5)$$P_{mix}=R_{p}\left(P_{max}\right)+R_{p}\left(P_{avg}-P_{max}\right)$$
where $P_{mix}$denotes the feature vector obtained after mixed pooling, $P_{Avg}$ and $P_{max}$ denote the feature vectors obtained after average pooling and maximum pooling, respectively, and $R_{P}$ denotes a 1 × 1 convolution operation on the feature map followed by *BN* and Relu activation.

### 2.4. Cross-Consistent Learning Module

In order to empower the feature encoder to extract stable features of person images, it is proposed to game different sets of samples crossed with classifiers equipped with specific knowledge. Firstly, the person image samples were randomly divided into two parts according to camera views; assuming that the dataset samples come from cameras with N views, one group contains [N/2] camera samples with one viewpoint and the other group contains N-[N/2] camera samples with one viewpoint down, where [ ] denotes the downward rounding function. Different camera views can be interpreted specifically as different camera angles. For example, the Market-1501 dataset has six cameras, which were randomly divided into two groups of three cameras each, and the DukeMTMC-ReID dataset has eight cameras, which were randomly divided into two groups of four cameras each.

As shown in Figure 5, a distinct classifier was trained for the feature vectors generated from person images from various sample groups. Given a person image feature vector f and crossing two distinct sets of classifiers, if the two distinct classifiers produce the same prediction, the network has successfully eliminated the impact of various camera perspective modifications.

The loss of the classifier cross-entropy function in the pre-training phase is Equation (6):(6)$$L_{CE}=-\frac{1}{n}\sum_{i=1}^{n}\sum_{p=1}^{P}-I\left(c,y_{1}^{i}\right)\log P\left(C_{1}\left(f_{1}^{i}|w_{1}\right)\right)-\frac{1}{m}\sum_{j=1}^{m}\sum_{p=1}^{P}-I\left(c,y_{2}^{j}\right)\log P\left(C_{2}\left(f_{2}^{j}|w_{2}\right)\right)\;$$
$$I\left(c,y\right)=\left\{\begin{array}{c} 1-\frac{N-1}{N}\varepsilon ,if\;c=y\\ \frac{\varepsilon }{N},otherwise \end{array}\right.$$
where *n* and *m* denote the total number of images in each batch of the two sample groups, respectively, *p* denotes the total number of person categories in the training set, $y_{1}^{i}$ denotes the true label corresponding to the training set image $x_{i}$, 1 denotes the first camera sample set, $y_{2}^{j}$ denotes the true label corresponding to the training set image $x_{j}$, 2 denotes the second camera sample set, $C_{1}\left(f_{1}^{i}|w_{1}\right)$ and $C_{2}\left(f_{1}^{i}|w_{2}\right)$ denote the classification probabilities obtained by classifier w**_1_** and w**_2_** for the images of the two camera views, respectively, *f_1_^i^* denotes the feature vector extracted from the images of camera sample group 1, *f_1_^j^* denotes the feature vector extracted from the images of camera sample group 2, and *c* is the predicted sample label, and the smoothing hyper-parameter ε takes a value of 0.1.

KL divergence is used to describe the similarity of two different probability distributions, and KL scatter possesses non-negativity, as shown in Equation (7).
(7)$$\sum_{i}P(i)ln\left(\frac{Q\left(i\right)}{P\left(i\right)}\right)\leq \sum_{i}P(i)\left(\frac{Q\left(i\right)}{P\left(i\right)}-1\right)=0\;$$

When and only when ∀i, $\frac{Q\left(i\right)}{P\left(i\right)}=1$ occurs, at which time *P* = *Q*, the equal sign is obtained. The proposed average camera view classifier, which has the same structure as the camera view classifier, enables consideration of not only the parameters of the current batch during the network update but also the parameters that are updated concerning the previous batch, with the crossover consistency loss shown in the equation. A direct crossover of classifiers with different knowledge would result in a rapid and optimized agreement of the two classifier parameters, Equation (8).
(8)$$L_{CCL}=\frac{1}{n}\sum_{i=1}^{n}C_{1}\left(f_{1}^{i}|E\left[w_{1}\right]\right)\log\frac{c_{1}\left(f_{1}^{i}|E\left[w_{1}\right]\right)}{c_{2}\left(f_{1}^{i}|w_{2}\right)}+\frac{1}{m}\sum_{j=1}^{m}C_{2}\left(f_{2}^{j}|E\left[w_{2}\right]\right)\log\frac{c_{2}\left(f_{2}^{j}|E\left[w_{2}\right]\right)}{c_{1}\left(f_{2}^{j}|w_{1}\right)}$$
where *E*[*w_1_*] and *E*[*w_2_*] denote the parameters of these two average classifiers, respectively, which are related to the batch of data set training and are updated after each batch of data training. The parameters of this set of average classifiers are initialized, *E*(0)[*w_1_*] = *w1* and *E*(0)[*w_2_*] = *w2*, and the average classifier parameters are updated as Equations (9) and (10).
(9)$$E^{\left(t\right)}\left[w_{1}\right]=\left(1-\beta \right)E^{\left(t-1\right)}\left[w_{1}\right]+\beta \left[w_{1}\right]$$
(10)$$E^{\left(t\right)}\left[w_{2}\right]=\left(1-\beta \right)E^{\left(t-1\right)}\left[w_{2}\right]+\beta \left[w_{2}\right]\;$$
where *E*(*t*)[*w_1_*] and *E*(*t* − *1*)[*w_1_*] denote the model parameters of the current and previous batches, respectively, and *β* is the equilibrium parameter of the model update, where *β* is set to 0.2.

The total loss of cross-consistency learning is Equation (11):(11)$$L_{S\mathrm{UM}}=L_{CE}+\mu L_{CCL}$$

During the pre-training phase, the cross-consistency loss balance parameter *μ* is set to 1.5, and the joint cross-entropy loss and cross-consistency learning loss of the two classifiers are jointly updated for the backbone network.

### 2.5. Feature Pyramid Module

A Feature Pyramid Network (FPN) [21] was initially used in object detection. Figure 6 depicts its structure.

In this paper, the pedestrian feature maps generated by different layers of the network are denoted by $T_{k}$($T_{k}$ ∈ T), and the semantic information contained in *T*_1_ and *T*_2_ is not considered as incomplete. The sizes of the extracted feature maps *T_3_* to *T_5_* have the following sizes and order: *T*_3_ ∈ R^512×32×16^, *T*_4_ ∈ R^1024×16×8^ and *T*_5_ ∈ R^2048×16×8^. Since the convolutional features of different sizes have to be fused accordingly, the convolutional kernels of size 1 × 1 are used to adjust the features of different levels to the same number of channels, and a feature map of uniform spatial dimension is obtained after the up-sampling operation of the features using Bilinear Interpolation (BII) [22]. Finally a final feature mapping is generated using a 3 × 3 convolution in each feature matrix incorporating multiple layers of features. The feature pyramid is calculated as shown in Equations (12)–(14).
(12)$$T_{i}^{\prime }=GAP\left(conv_{1\times 1}\left(T_{i}\right)\right)\;\;\;i=5\;$$
(13)$$T_{i}^{\prime }=GAP\left(conv_{3\times 3}\left(conv_{1\times 1}\left(T_{i}\right)\right)\right.⊕\left.\left(conv_{1\times 1}\left(T_{i+1}\right)\right)\right)\;i=4\;$$
(14)$$T_{i}^{\prime }=GAP\left(conv_{3\times 3}\left(conv_{1\times 1}\left(T_{i}\right)\right)\right.⊕blinear\left.\left(conv_{1\times 1}\left(T_{i+1}\right)\right)\right)\;i=3\;$$
where *conv* stands for convolution operation and bilinear stands for bilinear interpolation operation. Global Average Pooling (GAP) stands for global average pooling operation, and ⊕ stands for element-by-element summation operation. Following the feature pyramid module, three feature vectors are obtained. The feature vector ${T^{\prime }}_{5}$ is 512-dimensional after convolution and pooling procedures, while ${T^{\prime }}_{4}$ and ${T^{\prime }}_{3}$ are 256-dimensional due to the fifth layer of the feature map’s rich semantic meaning. The model is optimized using the same set of losses after the three feature vectors are combined into a 1024-dimensional vector in the channel dimension.

## 3. Loss Function

The person re-identification model was jointly trained using cosine cross-entropy loss, triplet loss, and cluster center loss improved on the cross-entropy loss function to jointly constrain the features extracted by the network model and optimize the proposed model.

### 3.1. Cross-Entropy Loss and Triplet Loss

In the field of deep learning-based pedestrian re-identification, most of the work uses cross-entropy loss in conjunction with triplet loss to train the network together. The cross-entropy loss function for label smoothing is shown in Equation (15).
(15)$$L_{CE}=-\sum_{i=1}^{N}q_{i}log\left(p_{i}\right)$$
$$p_{i}=\frac{exp\left(W_{y_{i}}^{T}f_{i}+b_{y_{i}}\right)}{\sum_{K=1}^{C}exp\left(W_{K}^{T}f_{i}+b_{K}\right)}q_{i}=\left\{\begin{array}{c} 1-\frac{N-1}{N},if\;i=y\\ \frac{\varepsilon }{N},otherwise \end{array}\right.$$
where *N* represents the total number of person images in each batch, *C* represents the total number of person identities in each batch, *f_i_* represents the feature vector of the image with the true label $y_{i}$, *W* represents the weight vector, and b is the bias value.

Difficult sample triplet loss is a widely used loss function in the field of image identification, where $f_{a}$, $f_{p}$, and $f_{n}$ denote Anchor pedestrian image features, pedestrian image features with the same label as the anchor, and pedestrian image features with a different label than the anchor, respectively, then the discrepancy between $f_{a}$, $f_{p}$ and $f_{a}$, $f_{n}$ is calculated separately, letting the difference in distance between the two be the parameter of the loss function. In order to make the algorithm more accurate during training, $f_{p}$ which is the farthest away from $f_{a}$, and $f_{n}$_,_ which is the closest distance from fa, are used as training data, and the triplet loss function is shown in Equation (16).
(16)$$L_{Triplet}=-\sum_{i=1}^{P}\sum_{a=1}^{K}[\alpha +\underset{p=1\cdot \cdot \cdot K}{max}{\left\|f_{a}^{\left(i\right)}-f_{p}^{\left(i\right)}\right\|}_{2}-\underset{\begin{array}{l} n=1\cdot \cdot \cdot K\\ j=1\cdot \cdot \cdot P\\ j\neq i \end{array}}{min}{\left\|f_{a}^{\left(i\right)}-f_{n}^{\left(j\right)}\right\|}_{2}{]}_{+}$$
where *P* denotes that there are images of pedestrians with different identities in the same batch of training data, *K* denotes that there are *K* instances of each identity of pedestrians in the same batch of training data, and α denotes the interval distance between positive and negative samples, where *α* = 0.3.

### 3.2. Cosine Softmax Loss

The cross-entropy loss function uses the inner product of vectors to measure similarity, while the triplet loss optimizes the Euclidean distance between the feature vectors. The Cosine Softmax Loss is different from the traditional Softmax Loss in that it optimizes the inner product between the sample vector and the weight vector, while the Cosine Softmax Loss optimizes the cosine distance between the sample vector and the weight vector.
(17)$$L_{COS}=\frac{1}{N}\sum_{i=1}^{N}-log\frac{e^{\gamma \left(s\left(W_{y_{i}}^{T},x_{i}\right)-m\right)}}{e^{\gamma \left(s\left(W_{y_{i}}^{T},x_{i}\right)-m\right)}+\sum_{\begin{array}{l} j=1\\ j\neq y_{i} \end{array}}^{C}e^{\gamma \left(s\left(W_{y_{i}}^{T},x_{i}\right)\right)}}$$
where *s*(*x*, *y*) calculates the cosine distance between *x* and *y*. The parameter m is the cosine interval that allows the network to make the feature vectors of the same person image less distant when optimized, and γ is the scale factor.

### 3.3. Loss of Cluster Center

To increase the cohesiveness of features within the same identity image, cluster center loss is proposed and combined with Cosine Softmax Loss and triplet loss to optimize the model, with cluster center loss regulating the relationship between features of different identity classes, as shown in Figure 7.

The average characteristics of the same identity are shown in Equation (18)
(18)$$h_{y_{i}}=\frac{\sum_{k=1}^{K}f_{k}}{K}$$
where Kdenotes the average feature value of the pedestrian image labeled $y_{i}$ and *K* is the number of instances of the same identity in a batch of training data.

To make the model converge faster during the training process, only the most difficult samples are considered during the training process, reducing the number of iterations required for the convergence process. The maximum Euclidean distance between the same identity pedestrian and the average feature of its corresponding identity is shown as Equation (19)
(19)$$d_{i}^{k\_max}=\underset{k\in K}{max}{\left\|f\left(x_{k}\right)-h_{y_{i}}\right\|}_{2}^{2}$$

The minimum Euclidean distance of the average intra-row person trait between different identities is shown as Equation (20)
(20)$$d_{i}^{p\_min}=\underset{\forall j\in p,j\neq i}{min}{\left\|h_{y_{i}}-h_{y_{j}}\right\|}_{2}^{2}$$
where *p* denotes the total number of personal identities in the same batch of data. During the iteration of each batch of data, the intra-class distance is reduced while the inter-class distance is increased, and the loss of cluster centers based on hard sample mining is shown as Equation (21)
(21)$$L_{Center}=\frac{1}{N}\sum_{\begin{array}{l} j=1\\ y_{i}\in P \end{array}}^{N}{\left\|f_{j}-h_{y_{i}}\right\|}_{2}^{2}+\frac{1}{P}\sum_{p=1}^{P}{\left[\rho +\left(d_{i}^{k\_max}-d_{i}^{p\_min}\right)\right]}_{+}$$

The total loss function in the multi-feature fusion stage is shown in Equation (22)
(22)$$L_{Total}=L_{COS}+L_{Triplet}+L_{CC}$$

## 4. Experiments and Analysis

The algorithms in this paper were implemented based on the pytorch framework, using a 64-bit Ubuntu 16.04 based computing platform with the following hardware configuration: NVIDIA GeForce GTX 3090 GPU and 64 GB of memory. The model was pre-trained in the first 50 iterations in the cross-consistency learning phase, and fine-tuned in the second 150 iterations in the multi-feature fusion phase, using the Adam optimizer to optimize the model parameters. During the training process, the training technique of REA [23] was introduced to randomly mask the person images, setting the probability of randomly erasing the images to 0.5 and the area ratio of the erased part to 0.02 < S < 0.4, as shown in Figure 8, which can reduce the degree of overfitting of the model.

### 4.1. Experimental Data Set

This paper conducted corresponding comparative experiments on the proposed person re-identification method on the currently more commonly used datasets Market-1501 and DukeMTMC-ReID, the details of which are shown in Table 1.

### 4.2. Comparison with Existing Methods

In this paper, the current widely used datasets Market-1501 and DukeMTMC-ReID were compared with the existing state-of-the-art methods and the results are shown in Table 2.

Although it somewhat increases the model’s identification rate, the key point detection model PIE provided in the table has alignment issues and uses more processing resources. The rank1 and mAP of the method in this paper were also higher than those of the PCB model using only local features and the PCB-U + RPP model after sampling fine-grained adjustment, respectively. Although AlignedReID incorporates global features based on dynamically aligned local features, it neglects the contribution of shallow features to image features, and therefore the recognition rate is also inferior to that of the CCLM model. The Attentive but Diverse Network (ABD-Net) model was only slightly higher in mAP than the CCLM model on the Market-1501 dataset, although the ABD-Net model uses automatic differentiation to calculate the spectral value difference orthogonality for training, a process with a large computational burden of feature decomposition, which increases the difficulty of training.

### 4.3. Analysis of Visualization Results

The ResNet50 residual network is the foundation of the proposed CCLM model. The baseline and CCLM model validation and analysis findings were visualized in this study in order to visually check the model’s validity. To match the four-person images in Figure 9 in the query set, four typical types of photographs—normal walking, cycling, obscured, and poor resolution—were arbitrarily chosen from the query set. Both the baseline activation map and the CCLM model activation map are represented by the matching heat maps of each image, respectively.

Figure 9 shows in (1b) that the baseline activation area in the image is concentrated in the legs, covering less of the rest of the body. (1c) shows that the CCLM model activation area covers more of the body area, enhancing the correlation between the various parts of the body. (2b) shows that the activation area is not only concentrated in the body parts of the person, but also the backpack area is covered, while in (2c) only the body parts are covered, mitigating the impact of the pose transformation. The activation area in (3b) also focuses on the handbag, where the baseline model identifies the pedestrian shorts in the wrong dataset as a handbag, and in (3c) it mainly covers the body part, reducing the interference of occlusions. (4b) and (4c) show that for the low-resolution images, the activation area of the CCLM model is also significantly improved over the baseline, validating the CCLM model in the query dataset validity. Where(1a), (2a), (3a) and (4a) are the original images.

Figure 10 shows the query results for the corresponding four person images, where the top row is the set of query results for the baseline and the bottom row is the set of query results for the CCLM model. The solid border indicates a correct query result and the dashed line against a red background indicates an incorrect query result.

The images of the same person are not a single scene in the gallery set. To further verify the validity of the CCLM model, four sets of person images were randomly selected for visual validation. As shown in Figure 11, the four groups of person images include four sets of comparison images of the person-unaligned, pose-transformed, person-occluded, and multiple-person noise respectively. In the person-unaligned and person-pose-transformed image groups, the area covered by the CCLM model was concentrated on the whole human body part. In the images with occlusion and multiple-person noises, the activation area of the CCLM model avoided the occlusion and other personal information, which effectively reduced the noise interference brought by other information and validates the effectiveness of the CCLM model in multiple data sets.

### 4.4. Ablation Experiments

To verify the effectiveness of each component of the proposed person re-identification model, ablation experiments for the corresponding modules were designed on two datasets, Market-1501 and DukeMTMC-ReID, using a single query model. The results of Rank-1 and mAP for each metric are shown in Table 3. CBAM denotes the Attention Mechanism Module, MPM denotes Mixed Pooling Module, FPN denotes Feature Pyramid Module, and CCL denotes Cross Consistency Learning Module. Only one set was changed for all results, and the rest of the settings were the same as the default settings.

#### 4.4.1. Impact of Different Modules

The CBAM, MPM, FPN, and CCL modules make up the majority of the CCLM model, and Table 3 displays the effects of the various modules on the experimental outcomes. The heat map of each image corresponds to the order of the ablation experiment in Table 3 from left to right, and Figure 12 displays the visualization features of four randomly chosen person images from the ablation experiment.

With the CBAM module fused in the baseline, the recognition rates of the model are shown in Table 3 to have improved in both datasets, and the activation regions are seen in Figure 12 to be more focused on the pedestrian parts rather than the image background. This is because the attention mechanism module allows the model to adaptively focus on the more discriminative features of the image. By replacing the mean pooling module with the MPM module on top of the embedded CBAM, the recognition rate was further improved because the MPM module focuses on the global features of the image as well as the texture features of the image, attenuating the effect of the background on feature extraction. Based on the abovementioned operation, the FPN module was embedded into the model, and it was evident in the heat map that the activation area of the visualization map of the four images covered almost the whole body part, and the table also showed a significant increase in recognition rate, which was due to the fact that not only deep features but also the contribution of shallow features to feature extraction were considered after feature fusion. In the case of pre-training the model with cross-consistency learning, the table also showed an improvement compared with the case without pre-training, since the influence of different camera views on feature extraction is reduced during the pre-training phase of the model, which enhances the robustness of the model.

#### 4.4.2. Effect of Different Loss Functions

Based on the cross-consistency pre-training of the model, in order to test the effect of the loss function on the recognition rate of the model, corresponding experimental validations were conducted on the Market-1501 and DukeMTMC-ReID datasets, respectively, and the experimental results are shown in Table 4.

Where Cos denotes the Cosine Softmax Loss, Triplet denotes the triplet loss and Center denotes the cluster center loss. The results showed that the worst results are obtained when one loss function is used, and the best results are obtained when the model is trained with the combination of the three loss functions, and the experiments validate the effectiveness of the loss functions used.

#### 4.4.3. The Impact of the REA Module

As can be seen from Table 5, the Rank-1 in CCLM without the REA module was lower than the experimental results with the REA module, because REA reduces the interference of background noise and focuses on the part of the pedestrian rather than learning the whole object.

#### 4.4.4. Selection of Parameters

The parameters γ and m of Cosine Softmax Loss and the parameter ρ of cluster center loss were experimentally confirmed on the Market-1501 dataset to show the reasonableness of the parameters. The parameters are shown in Figure 13.

The parameter γ is the scale factor in the Cosine Softmax loss function, Rank-1 and mAP have the best performance index when γ is 60. The parameter *m* is the interval in the Cosine Softmax loss function, which is used to strengthen the cohesion of the image features, *m* was set to 0.3 to achieve the highest recognition rate. Although the highest Rank-1 metric was achieved for both ρ values of 0.6 and 0.8, the mAP metric was higher for *ρ* values of 0.8 and was therefore set to 0.8 in this paper.

### 4.5. Results of the Actual Scenario

In order to verify the effectiveness and practicality of the proposed model, this model was combined with the YOLO V3 target detection model to retrieve specific persons in two different real-world scenarios.

The experiment confirms that the model in this study has good feature extraction capabilities because it can still identify the persons to be retrieved in the presence of occlusions and significant changes in human position, as shown in Figure 14 and Figure 15.

## 5. Conclusions

In the training process, the proposed person re-identification model based on cross-consistency learning and multi-feature fusion was divided into two phases: model pre-training and model fine-tuning. The cross-consistency module of the model suppressed noise interference from non-person-identified semantic information during the pre-training stage. In the fine-tuning stage of the model, different spatial semantic relationships were established on multiple scales of feature information to enhance the model’s representation of the semantic information of the images. The effectiveness of the proposed model was validated in several datasets as well as in real-world scenes, improving the model’s ability to extract image detail features. The proposed network model’s first accuracy on the datasets Market-1501 and DukeMTMC-ReID was 95.9% and 89.7%, respectively. The main focus of future work will be on applying models to real-world scenarios, and lightweight models in conjunction with knowledge distillation will be the next major step.

## Figures and Tables

**Figure 1 sensors-22-09387-f001:**
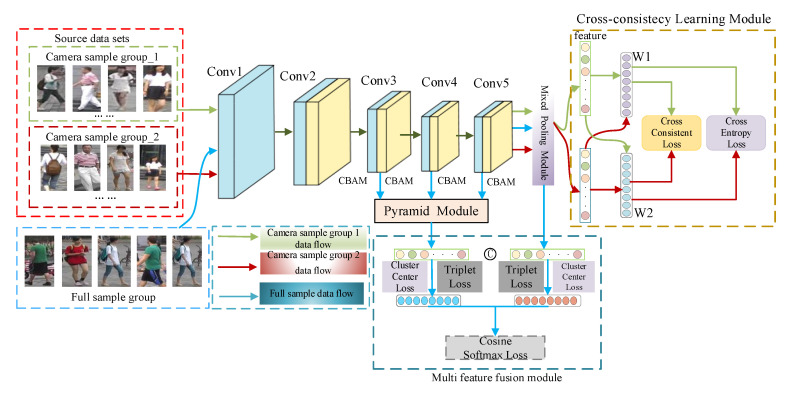
General model structure diagram.

**Figure 2 sensors-22-09387-f002:**
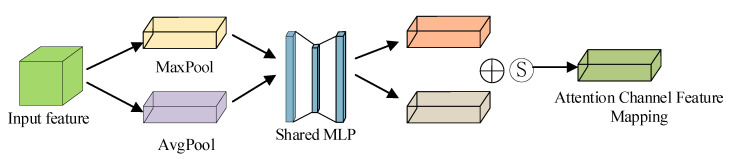
Channel attention mechanism.

**Figure 3 sensors-22-09387-f003:**
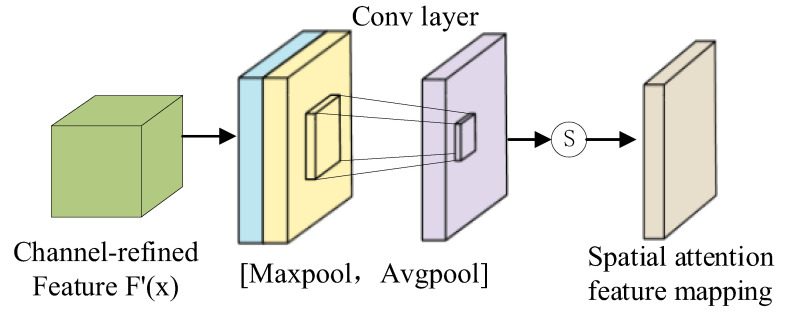
Spatial attention mechanism.

**Figure 4 sensors-22-09387-f004:**
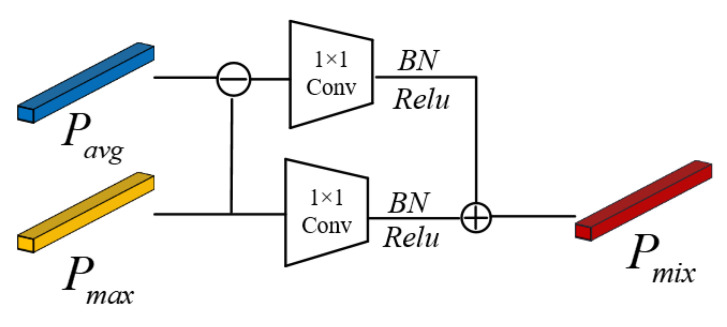
Mixed-pooling module.

**Figure 5 sensors-22-09387-f005:**
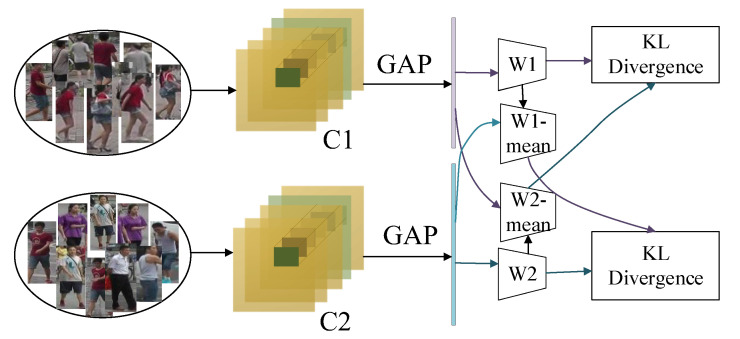
Cross consistency learning module.

**Figure 6 sensors-22-09387-f006:**
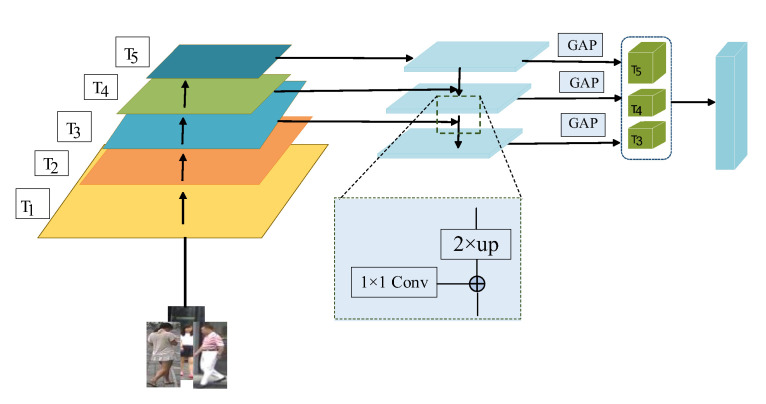
Feature pyramid module diagram.

**Figure 7 sensors-22-09387-f007:**
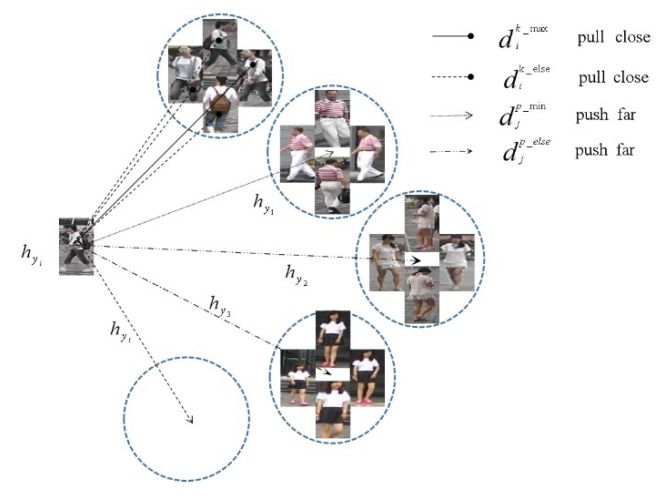
Cluster center visualization.

**Figure 8 sensors-22-09387-f008:**
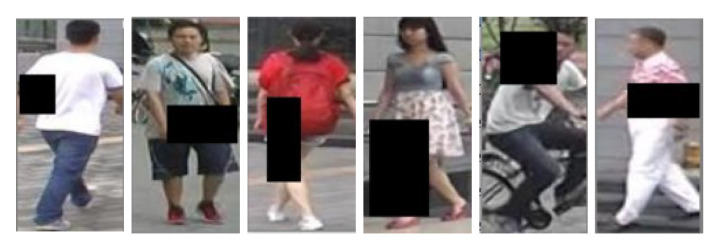
Schematic diagram of random occlusion.

**Figure 9 sensors-22-09387-f009:**
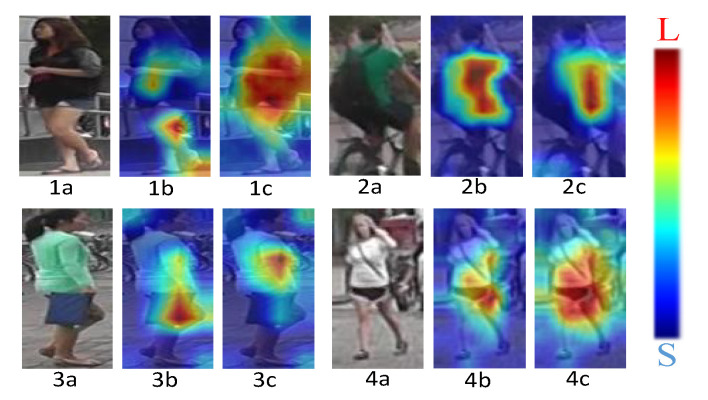
Visual thermal map for persons.

**Figure 10 sensors-22-09387-f010:**
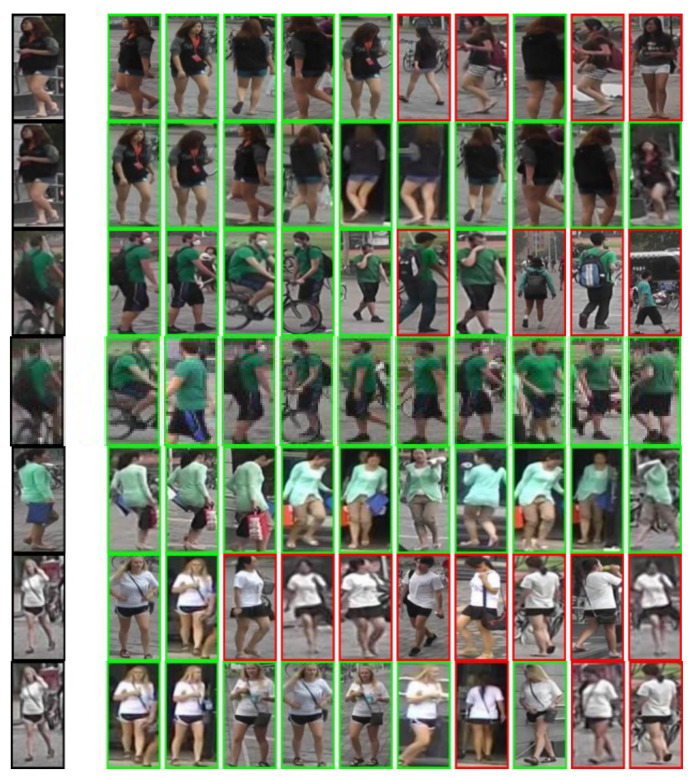
Schematic diagram of person query results.

**Figure 11 sensors-22-09387-f011:**
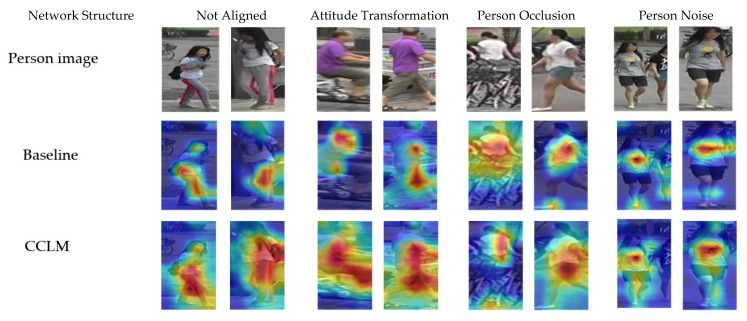
Visual thermal map of scene transformation.

**Figure 12 sensors-22-09387-f012:**
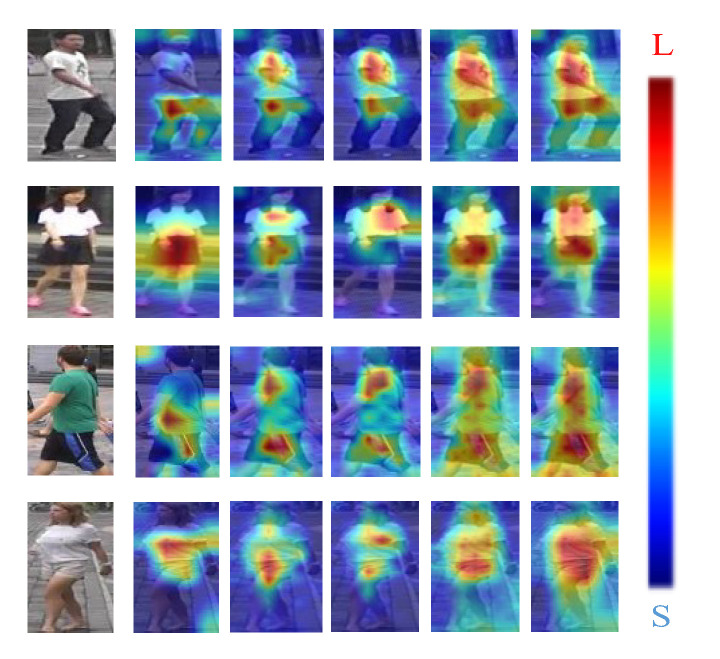
Visualized thermal map of ablation experiments.

**Figure 13 sensors-22-09387-f013:**
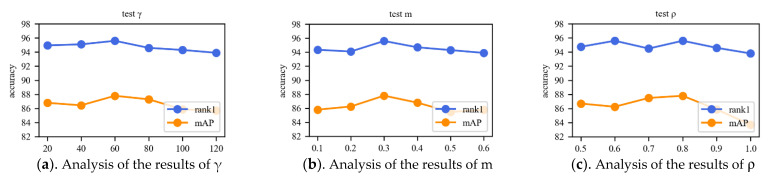
Results of different parameters.

**Figure 14 sensors-22-09387-f014:**
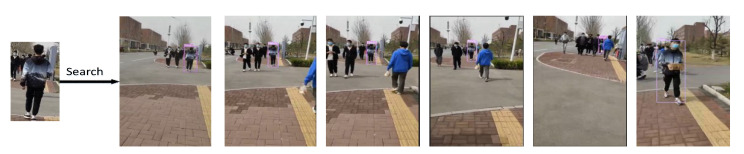
Detection result of the scene picture taken by mobile phone.

**Figure 15 sensors-22-09387-f015:**
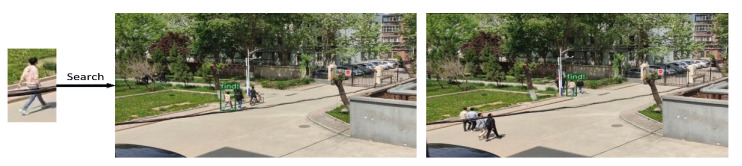
Search result of the monitoring shooting scene.

**Table 1 sensors-22-09387-t001:** Attribute information of the dataset.

Dataset	Market-1501	DukeMTMC-ReID
ID	1501	1404
Camera	6	8
Image	32,668	36,411
Train ID	750	702
Test ID	751	702
Retrieved Image	3368	2228
Candidate Set Image	19,732	17,661

**Table 2 sensors-22-09387-t002:** Different methods that were used to evaluate the results.

Method	Market-1501		DukeMTMC-ReID	
	Rank-1	mAP	Rank-1	mAP
XQDA [6] (Natural Science Edition 2018)	43.0	21.7	31.2	17.2
IDE [8] (ArxiV 2016)	72.5	46.0	65.2	44.9
PCB [11] (ECCV 2018)	92.4	77.3	81.9	65.3
PIE [13] (IEEE 2019)	78.7	53.9	—	—
AlignedReID [13] (2019)	90.6	77.7	81.2	67.4
ABD-Net [17] (IEEE 2019)	95.6	88.2	89.0	78.5
SVDNet [24] (IEEE 2017)	82.3	62.1	76.7	56.8
ABFE-Net [25] (2022)	94.4	85.9	88.3	75.1
HA-CNN [26] (IEEE 2017)	91.2	75.7	80.5	63.8
PCB-U+RPP [27] (IEEE 2019)	93.8	81.6	84.5	71.5
CASN(PCB) [28] (IEEE 2019)	94.4	82.8	87.7	73.7
HOReID [29] (IEEE 2020)	94.2	84.9	86.9	75.6
SNR [30] (IEEE 2020)	94.4	84.7	84.4	72.9
TSNet [31] (2021)	94.7	86.3	86.0	75.0
BDB [32] (IEEE 2019)	95.3	86.7	89.0	76.0
SCAL(spatial) [33] (ICCV 2019)	95.4	88.9	89.0	79.6
SCAL (channel) [33] (ICCV 2019)	95.8	89.3	88.9	79.1
RGA-SC [34] (CVPR 2020)	96.1	88.4	—	—
TransReID [35] (ICCV 2021)	95.2	89.5	90.7	82.6
MGN+UP-ReID [36] (CVPR 2022)	97.1	91.1	—	—
**CCLM (Ours)**	**95.9**	**88.1**	**89.7**	**79.3**

**Table 3 sensors-22-09387-t003:** Ablation experiment data sheet.

Network Structure	Market-1501		DukeMTMC-ReID	
	Rank-1	mAP	Rank-1	mAP
Baseline	89.3	74.5	75.2	62.5
Baseline + CBAM	92.2	76.4	81.5	65.4
Baseline + CBAM + MPM	92.4	78.9	82.3	69.2
Baseline + CBAM + MPM + FPN	94.4	85.5	86.4	73.4
Baseline + CBAM + MPM + FPN + CCL	**95.9**	**88.1**	**89.7**	**79.3**

**Table 4 sensors-22-09387-t004:** Table of results of different loss functions.

Loss Function	Market-1501		DukeMTMC-ReID	
	Rank-1	mAP	Rank-1	mAP
Cos	93.4	84.7	86.3	76.5
Cos + Triplet	94.2	85.6	88.4	77.9
Cos + Triplet + Center	**95.9**	**88.1**	**89.7**	**79.3**

**Table 5 sensors-22-09387-t005:** REA ablation experiment data sheet.

Network Structure	Market-1501		DukeMTMC-ReID	
	Rank-1	mAP	Rank-1	mAP
CCLM	93.5	86.2	87.3	74.6
CCLM + REA	95.9	88.1	89.1	79.3

## Data Availability

The datasets used are all publicly available datasets. The Market1501 dataset is available at: https://github.com/zhunzhong07/IDE-baseline-Market-1501. The DukeMTMC-reID dataset is available at: https://github.com/RenMin1991/cleaned-DukeMTMC-reID.

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
