# Peer review of "Joint Cross-Consistency Learning and Multi-Feature Fusion for Person Re-Identification"

_sensors, 2022, doi:10.3390/s22239387_

Round 1
Reviewer 1 Report
Strengths:
1. This paper addressed the problem of person re-identification which is an active topic in computer vision.
2. The structure of the paper is coherent. The method was precisely described as well as the results were clearly presented.
3. The amount of works and experiments in this paper is quite large and the authors provided many significantly results and comparisons.
Weaknesses:
1. The novelty/originality of this paper is just average. The idea of using multi-level features fusion to improve performance were existing in many previous researches.
2. There are several spell and typo errors.
Suggestions to improve:
2. The paper needs some spell and typo checks: line 18, 29, 35, 42, 237-241, equation 16, etc.
2. The references [16-18] were not cited in the content paper.
3. The publishing year of the compared methods should be added in Table 2.
4. The authors should compare their methods with more recent state-of-the-art metioned in following papers [1-4]
[1] Self-Critical Attention Learning for Person Re-Identification. ICCV 2019
[2] Relation-Aware Global Attention for Person Re-identification. CVPR 2020
[3] TransReID: Transformer-Based Object Re-Identification. ICCV 2021
[4] Unleashing Potential of Unsupervised Pre-Training with Intra-Identity
Regularization for Person Re-Identification. CVPR 2022
Author Response
Response to Reviewer 1 Comments
Dear reviewers
Re: Manuscript ID: 1999662 and Title: Joint cross-consistency learning and multi-feature fusion for person re-identification
Thank you for your letter and the reviewers’ comments concerning our manuscript entitled “Joint cross-consistency learning and multi-feature fusion for person re-identification” (1999662). Those comments are valuable and very helpful. We have read through comments carefully and have made corrections. Based on the instructions provided in your letter, we uploaded the file of the revised manuscript. Revisions in the text are shown using red highlight for additions, and strikethrough font for deletions.
We would love to thank you for allowing us to resubmit a revised copy of the manuscript and we highly appreciate your time and consideration.
Sincerely.
Author responses.
Here are our responses and any changes will be indicated in the revised article.
Point 1:The novelty/originality of this paper is just average. The idea of using multi-level features fusion to improve performance were existing in many previous researches
Reply:
To address the problems of the above methods, this paper investigates the extraction of pedestrian image features. We designed a model joint cross-consistency learning and multi-feature fusion for person re-identification model CCLM (Cross Consistent Learning and Multi-feature) for Person re-identification. The main work of this paper is as follows.
1) The ResNet50 residual network [18] is used as the backbone network, and the interference of invalid feature information is reduced by embedding the dual-attention mechanism module Convolutional Block Attention Module (CBAM) [19] and the mixed pooling module (MPM) in the residual network.
2) In the cross-consistency learning module of the model, the model is pre-trained to design image classifiers specific to different viewpoints to extract features from different viewpoints and reduce the impact of viewpoint transformation on the extraction of image information.
3) The model is fine-tuned in the multi-feature fusion module, and the fused multi-level features are used to match the similarity with images from the image library to enhance the feature representation capability of the model.
4) In the process of model optimization, for the difference of multiple losses in the optimization space, Cosine Softmax loss [20] is introduced to eliminate the spatial in-consistency of cross-entropy loss and triplet loss, and Cluster Center Loss (CCL) is proposed to make the model focus on intra-class distance as well as inter-class distance in the optimization process.
We have made changes in response to your comments. This change is in page 2, line 78-97. The innovations in this paper are as follows:
Point 2:There are several spell and typo errors. Suggestions to improve: The paper needs some spell and typo checks: line 18, 29, 35, 42, 237-241, equation 16, etc.
Reply:
page 1 - line 18 => "images are alleviated.Then the multi-level" -> "images are alleviated. Then the multi-level"
We have made changes in response to your comments. This change is in page 1-line 18.
Reply:
page 1 - line 29 => "deep learning methods.The traditional person" -> "deep learning methods. The traditional person"
We have made changes in response to your comments. This change is in page 1-line 29.
Reply:
page 1 - line 31 => "discriminative and recognisable person image" -> "discriminative and recognizable person image"
We have made changes in response to your comments. This change is in page 1-line 31.
Reply:
page 1 - line 33 => "by obtaining colour information of "-> "by obtaining color information of"
We have made changes in response to your comments. This change is in page 1-line 33.
Reply:
page 1 - line 35 => "of features.The aim of "-> "of features. The aim of"
We have made changes in response to your comments. This change is in page 1-line 35.
Reply:
page 1 - line 42 => "Zheng et al.[9] proposed an I"-> " Zheng et al. [9] proposed an I"
We have made changes in response to your comments. This change is in page 2-line 53.
Reply:
page 2 - line 45 => "Hermans et al.[10] used a triplet"-> "Hermans et al. [10] used a triplet"
We have made changes in response to your comments. This change is in page 2-line 56.
Reply:
page 2 - line 49 => "Zheng et al.[12] proposed"-> "Zheng et al. [12] proposed"
We have made changes in response to your comments. This change is in page 2-line 61.
Reply: page 2 - line 52 => "Luo et al.[13] proposed"-> "Luo et al. [13] proposed"
We have made changes in response to your comments. This change is in page 2-line 63.
Reply: page 2 - line 54 => " the features.Zheng et al. [14] "-> " the features. Zheng et al. [14]"
We have made changes in response to your comments. This change is in page 2-line 66.
Reply:
page 2-line 56 => "model's generalisation performance."-> "model's generalization performance."
We have made changes in response to your comments. This change is in page 2-line 68.
Reply:
page 2 - line 59 => "other factors.To avoid"-> "other factors. To avoid"
We have made changes in response to your comments. This change is in page 2-line 73.
Reply:
page 2 - line 87 => "finally optimises the backbone network"-> "finally optimizes the backbone network"
We have made changes in response to your comments. This change is in page 3-line129.
Reply:
page 2 - line 93 => "module is depictes in Figure 2"-> "module is depicted in Figure 2"
We have made changes in response to your comments. This change is in page 3-line135.
Reply:
page 3 - line 105 => "where F(x) denotes the image "-> "Where F(x) denotes the image"
We have made changes in response to your comments. This change is in page 4-line 151.
Reply:
page 6 - line 218 => "operation, bilinear blinear stands"-> "operation, bilinear stands"
We have made changes in response to your comments. This change is in page 7-line 281.
Reply:
page 7 - line 237-241=> " where fa、fp and fn represent the target image, positive sample image and negative sample image respectively, then the discrepancy between fa、fp and fa、fn is calculated separately and let the difference in distance between the two be the parameter of the loss function. In order to make the algorithm more accurate during training, fp, which is the farthest away from fa, and fn,which is the closest distance from fa, are used as training data, and the triplet loss function is shown in equation (16)."->
"Where, ,denote Anchor pedestrian image features, pedestrian image features with the same label as Anchor, and pedestrian image features with a different label than Anchor, respectively where 、 and represent the target image, positive sample image and negative sample image respectively, then the discrepancy between、and、 is calculated separately and let the difference in distance between the two be the parameter of the loss function. In order to make the algorithm more accurate during training, which is the farthest away from, and ,which is the closest distance from , are used as training data, and the triplet loss function is shown in equation (16)."
We have made changes in response to your comments. This change is in page 8-line 307-314.
The references [16-18] were not cited in the content paper.
Reply:
The references [16-18] have been added to the contribution
We have made changes in response to your comments. References [18-20] replace references [16-18] due to the inclusion of other scholars' work in the introduction. This change is in page 2-line 78-97.
Point 3:The publishing year of the compared methods should be added in Table 2.
Reply:
We have made changes in response to your comments. After reviewing the corresponding articles, the corresponding years have been written out in ()
Tab.2 Different methods were used to evaluate the results
Method |
Market -1501 |
DukeMTMC-ReID |
rank-1 mAP |
rank-1 mAP |
|
XQDA[6]( Natural Science Edition 2018) |
43.0 21.7 |
31.2 17.2 |
IDE[9] (ArxiV 2016) |
72.5 46.0 |
65.2 44.9 |
PIE[14] (IEEE 2019) |
78.7 53.9 |
— — |
SVDNet[24] (IEEE 2017) |
82.3 62.1 |
76.7 56.8 |
AlignedReID[13](2018) |
90.6 77.7 |
81.2 67.4 |
HA-CNN[26] (IEEE 2017) |
91.2 75.7 |
80.5 63.8 |
PCB[11] (ECCV 2018) |
92.4 77.3 |
81.9 65.3 |
ABFE-Net[25] (2022) |
94.4 85.9 |
88.3 75.1 |
PCB-U+RPP[27] (IEEE 2019) |
93.8 81.6 |
84.5 71.5 |
CASN(PCB)[28] (IEEE 2019) |
94.4 82.8 |
87.7 73.7 |
HOReID[29] (IEEE 2020) |
94.2 84.9 |
86.9 75.6 |
SNR[30] (IEEE 2020) |
94.4 84.7 |
84.4 72.9 |
TSNet[31] (2021) |
94.7 86.3 |
86.0 75.0 |
BDB[32] (IEEE 2019) |
95.3 86.7 |
89.0 76.0 |
ABD-Net[17] (IEEE 2019) |
95.6 88.2 |
89.0 78.5 |
SCAL(spatial)[33] (ICCV 2019) |
95.4 88.9 |
89.0 79.6 |
SCAL (channel)[33] (ICCV 2019) |
95.8 89.3 |
88.9 79.1 |
RGA-SC[34] (CVPR 2020) |
96.1 88.4 |
— — |
TransReID[35] (ICCV 2021) |
95.2 89.5 |
90.7 82.6 |
MGN+UP-ReID[36] (CVPR 2022) |
97.1 91.1 |
— — |
CCLM(Ours) |
95.9 88.1 |
89.7 79.3 |
Point 4:The authors should compare their methods with more recent state-of-the-art metioned in following papers [1-4]
[1] Self-Critical Attention Learning for Person Re-Identification. ICCV 2019
[2] Relation-Aware Global Attention for Person Re-identification. CVPR 2020
[3] TransReID: Transformer-Based Object Re-Identification. ICCV 2021
[4]Unleashing Potential of Unsupervised Pre-Training with Intra-Identity Regularization for Person Re-Identification. CVPR 2022
Reply:
We have made changes in response to your comments. Please see Table 2 for specific comparison experiments

Reviewer 2 Report
Please see the attachment

Author Response
Response to Reviewer 2 Comments
Dear reviewers
Re: Manuscript ID: 1999662 and Title: Joint cross-consistency learning and multi-feature fusion for person re-identification
Thank you for your letter and the reviewers’ comments concerning our manuscript entitled “Joint cross-consistency learning and multi-feature fusion for person re-identification” (1999662). Those comments are valuable and very helpful. We have read through comments carefully and have made corrections. Based on the instructions provided in your letter, we uploaded the file of the revised manuscript. Revisions in the text are shown using red highlight for additions, and strikethrough font for deletions.
We would love to thank you for allowing us to resubmit a revised copy of the manuscript and we highly appreciate your time and consideration.
Sincerely.
Author responses.
Here are our responses and any changes will be indicated in the revised article.
Point 1:In general, this paper is very purposeful. All the work in the paper is aimed at a very precise target. However, the reliability and stability of proposed methods is still doubted. In addition, the details of English writing and illustrations should be paid attention to.
Reply:
After calculating the mean of the experiments and proving experimentally on the Market501 and Duke MTMC-ReID datasets, the method used in this paper is more stable and the writing problems in the article have been carefully checked and corrected and the graphs have been revised.
Point 2:Page 3, line 107. and1 ∈ R c×c/r or and W1 ∈ R c×c/r? The latter seems more appropriate.
Reply:
We have made changes in response to your comments:
CBAM is a simple and effective attention module for feed-forward convolutional neural networks. Given an intermediate feature map, the module will infer the attention map sequentially along two separate dimensions (channel and space) and then multiply the attention map by the input feature map for adaptive feature extraction.
The process is as follows.
The input feature map F H×W×C is subjected to width and height-based Global Max Pooling and Global Average Pooling to obtain two 1×1×C feature maps, which are then fed into a two-layer Multilayer Perceptron (MLP). The number of neurons in the first layer is C/r (r is the reduction rate) and the activation function is Relu; the number of neurons in the second layer is C. This two-layer neural network is shared. Afterward, the MLP output features are subjected to element-wise summation operation and then sigmoid activation operation to generate the final channel attention feature, M_c. Finally, M_c and the input feature map F are subjected to element-wise multiplication operation to generate the spatial attention module.
The above response is quoted from Reference. [19]
[19] Woo, S.; Park, J.; Lee, J. Y.; & Kweon, I. S. CBAM: Convolutional block attention module. In Proceedings of the European conference on computer vision (ECCV), (2018),(pp. 3-19).
Point 3: Tab.1, Tab.2, Tab.4.The row numbers were placed in the tables by mistake.
Please check the Latex code to fix it.
Reply:
Tab.1, Tab.2 and Tab.4. The row have been modified to correspond to the position of the rows.
Reply:
Tab.1 We have made changes in response to your comments. This change is in page 1-line 396-404
Tab.2 We have made changes in response to your comments. This change is in page 11-line 428-451
Tab.4 We have made changes in response to your comments. This change is in page 14-line 549-555
Point 4:Page 9, Tab.2.The table should be settled in the same page.
Reply:
We have made changes in response to your comments. This change is in page 11-line 428-451
Point 5:Page 8, Section4.1. During the training process, the training technique of REA was introduced. It is strongly suggested to add ablation experiments to prove whether the REA can reduce the degree of overfitting of the model or not several syntaxes and grammatical need to check, moreover, some sentences are not easy to understand to the reader. Seen line 100-103 page 3.
Reply:
4.5.3 The impact of the REA module
Tab.5 REA Ablation Experiment Data Sheet
Network structure |
Market -1501 |
DukeMTMC-ReID |
rank-1 mAP |
rank-1 mAP |
|
CCLM |
93.5 86.2 |
87.3 74.6 |
CCLM+REA |
95.9 88.1 |
89.1 79.3 |
As can be seen from Table 5, the Rank-1 in CCLM without the REA module is lower than the experimental results with the REA module, because REA reduces the inter-ference of background noise and focuses on the part of the pedestrian rather than learning the whole object.
We have made changes in response to your comments. This change is in page 14-line 561-570.

Reviewer 3 Report
This paper is proposing a model with a joint cross‐consistency learning and multi‐feature fusion person re‐recognition algorithms. the paper is well presented and discussed, however there is some drawbacks need to be improved in the revised version.
1- The abstract is not technical and needs to highlight the research gap clearly.
2- Several syntaxes and grammatical need to check, moreover, some sentences are not easy to understand to the reader. Seen line 100-103 page 3.
3- Abbreviation should be defined in the text, such as (PCB, CCLM, CBAM, BN).
4- Don't add heading over heading. Add a few lines related to the detail of a particular section before starting a sub-section. i.e. Don’t add a new section and subsection directly, see section 2 and 2.1., section 4.5 and 4.5.1.
5- At the end of the Introduction section, add the contributions clearly.
6- The literature needs more work to figure out the gap of this research area.
7- “The cross‐consistency learning module randomly divides the person re‐ identification dataset into two groups based on the camera view” the camera view is not clear to the reader; the authors need to declare the sentences.
8- What is y1 and y2 in equation 6.
9- The symbols inside the text should be in italic format.
10- where fa、fp and fn represent the target image, positive sample image and negative sample image respectively, is it positive and negative classified image? The sentences should be clear to understand.
Author Response
Response to Reviewer 3 Comments
Dear reviewers
Re: Manuscript ID: 1999662 and Title: Joint cross-consistency learning and multi-feature fusion for person re-identification
Thank you for your letter and the reviewers’ comments concerning our manuscript entitled “Joint cross-consistency learning and multi-feature fusion for person re-identification” (1999662). Those comments are valuable and very helpful. We have read through comments carefully and have made corrections. Based on the instructions provided in your letter, we uploaded the file of the revised manuscript. Revisions in the text are shown using red highlight for additions, and strikethrough font for deletions.
We would love to thank you for allowing us to resubmit a revised copy of the manuscript and we highly appreciate your time and consideration.
Sincerely.
Author responses.
Here are our responses and any changes will be indicated in the revised article.
Point 1:The abstract is not technical and needs to highlight the research gap clearly.
Reply:
page 1 - line 8-10 => "To address the problem of insufficient feature extraction caused by the influence of viewpoint, occlusion, pose, and backdrop elements in the person re-recognition challenge, this paper proposes a model with a joint cross-consistency learning and multi-feature fusion person re-recognition. "-> "To solve the problem of inadequate feature extraction by the model due to factors such as occlusion and illumination in person re-identification tasks. this paper proposes a model with a joint cross-consistency learning and multi-feature fusion person re-identification."
We have made changes in response to your comments. This change is in page1-line 8-10.
Point 2:Several syntaxes and grammatical need to check, moreover, some sentences are not easy to understand to the reader. Seen line 100-103 page 3.
Reply:
page 3 - line 100-103 => " After the Sigmoid activation operation, the feature map F(x) of the image x extracted in the backbone network is input to the maximum pooling and average pooling layers to obtain two feature vectors FMax(x) and FAvg(x), respectively. The two feature vectors are then input to a multi-layer perceptron with shared parameters, and finally the corresponding channel attention mapping matrix is generated. Channel domain is one-dimensional. The attention mask is calculated as equation (1):" ->
" The feature map F(x) of the image x extracted in the backbone network is input to the max pooling and average pooling layers to obtain two feature vectors and respectively, and then the two feature vectors are input to the multilayer perceptron with shared parameters, finally, the corresponding channel attention mapping matrix is generated after the Sigmoid activation operation. The one-dimensional attention mask for the channel domain is calculated as equation (1)".
We have made changes in response to your comments. This change is in page3-line 141-148.
Point 3:Abbreviation should be defined in the text, such as (PCB, CCLM, CBAM, BN).
Reply:
page 2 - line 47 => "person re-identification method PCB, in which" ->" person re-identification method, (Part-based Conv Baseline) PCB, in which"
We have made changes in response to your comments. This change is in page 2-line 59.
Reply:
page 2 - line 63 => "CCLM" -> " model CCLM (Cross Consistent Learning and Multi-feature) for Person re-identification."
We have made changes in response to your comments. This change is in page 2-line 80.
Reply:
page 2 - line 91 => " The CBAM module consists " -> " The Convolutional Block Attention Module (CBAM) module consists"
We have made changes in response to your comments. This change is in page 2-line 84.
Reply:
page 4 - line 138 => " through the BN and Relu" -> "through the Batch Normalization(BN) and Relu"
We have made changes in response to your comments. This change is in page 5-line 190.
Reply:
page 9 - line 319 => " The ABD-Net model is only > " The Attentive but Diverse Network (ABD-Net) model is only"
We have made changes in response to your comments. This change is in page 10-line 415.
Point 4:Don't add heading over heading. Add a few lines related to the detail of a particular section before starting a sub-section. i.e. Don’t add a new section and subsection directly, see section 2 and 2.1., section 4.5 and 4.5.1.
Reply:
Section 2 and 2.1
- Algorithm flow
In this section, the Joint cross-consistency learning and multi-feature fusion for person re-identification network model are presented in five parts, starting with the general structure of the network, followed by the attention mechanism used by the network, followed by the mixed pooling module; then the cross-consistency learning module of the network, and finally explaining the feature pyramid module of the net-work. The modules collaborate with each other to improve the ability of the model to extract features.
2.1 Network structure
In order to extract the image detail information CCLM person re-identification model based on the ResNet50 residual network,
We have made changes in response to your comments. This change is in page 2-line 101-108
Reply:
Section 3 and 3.1
- Loss function
The person re-identification model is jointly trained using cosine cross-entropy loss, triplet loss, and cluster center loss improved on the cross-entropy loss function to jointly constrain the features extracted by the network model and optimize the pro-posed model.
3.1 Cross-entropy loss and triplet loss
In the field of deep learning-based pedestrian re-identification,
We have made changes in response to your comments: This change is in page 8-line 294-297.
Reply:
Section 4 and 4.1
As heading 4 Experiments and analysis overlaps with heading 4.1, make the following changes: move the text of 4.1 to the text of 4 and delete the heading 4.1 Experimental environment
- Experiments and analysis
The algorithms in this paper were implemented based on the pytorch framework, using a 64-bit Ubuntu 16.04 based computing platform with the following hardware configuration: NVIDIA GeForce GTX 3090 GPU and 64GB of memory. The model was pre-trained in the first 50 iterations in the cross-consistency learning phase, and fi-ne-tuned in the second 150 iterations in the multi-feature fusion phase, using the Ad-am optimizer to optimize the model parameters. During the training process, the training technique of REA [21] was introduced to randomly mask the person images, setting the probability of randomly erasing the images to 0.5 and the area ratio of the erased part to 0.02<S<0.4, as shown in Figure 8, which can reduce the degree of overfit-ting of the model.
Fig.8 Schematic diagram of random occlusion
4.1 Experimental data set
This paper conducts corresponding comparative experiments on the proposed person re-identification method on the currently more commonly used datasets Mar-ket-1501 and DukeMTMC-ReID, the details of which are shown in Table 1.
We have made changes in response to your comments: This change is in page 10-line 380-392.
Reply:
Section 4.5 and 4.5.1
Heading 4.5 becomes 4.4 and heading 4.5.1 becomes 4.4.1 as a result of the deletion of heading 4.1
4.4 Ablation experiments
To verify the effectiveness of each component of the proposed person re-identification model, ablation experiments for the corresponding modules were de-signed on two datasets, Market1501 and DukeMTMC-ReID, using a single query model. The results of Rank-1 and mAP for each metric are shown in Table 3. CBAM denotes the Attention Mechanism Module, MPM denotes Mixed Pooling Module, FPN denotes Feature Pyramid Module, and CCL denotes Cross Consistency Learning Module. Only one set was changed for all results, and the rest of the settings were the same as the default settings.
4.4.1 Impact of different modules
The CBAM, MPM, FPN, and CCL modules make up the majority of the CCLM model, and Table 3 displays the effects of the various modules on the experimental outcomes.
We have made changes in response to your comments: This change is in page 13-line 510-516.
Point 5:At the end of the Introduction section, add the contributions clearly.
Reply:
Chen [17] combined both channel and spatial information, but the regions extracted by combining only attention networks lacked adequate semantic interpretation and could not cover all discriminative image feature information.
To address the problems of the above methods, this paper investigates the extraction of pedestrian image features. We designed a model joint cross-consistency learning and multi-feature fusion for person re-identification model CCLM (Cross Consistent Learning and Multi-feature) for Person re-identification. The main work of this paper is as follows.
1) The ResNet50 residual network [18] is used as the backbone network, and the interference of invalid feature information is reduced by embedding the dual-attention mechanism module Convolutional Block Attention Module (CBAM) [19] and the mixed pooling module (MPM) in the residual network.
2) In the cross-consistency learning module of the model, the model is pre-trained to design image classifiers specific to different viewpoints to extract features from different viewpoints and reduce the impact of viewpoint transformation on the extraction of image information.
3) The model is fine-tuned in the multi-feature fusion module, and the fused multi-level features are used to match the similarity with images from the image library to enhance the feature representation capability of the model.
4) In the process of model optimization, for the difference of multiple losses in the optimization space, Cosine Softmax loss [20] is introduced to eliminate the spatial in consistency of cross-entropy loss and triplet loss, and Cluster Center Loss (CCL) is proposed to make the model focus on intra-class distance as well as inter-class distance in the optimization process.
This paper proposes the CCLM, a joint cross-consistency learning and multi feature fusion person re-identification model, to address the drawbacks of the earlier approaches.
We have made changes in response to your comments: This contribution is in page 2-line 78-97.
Point 6:The literature needs more work to figure out the gap of this research area.
Reply:
To identify more gaps with other approaches in the field, two references have been added
Reference:
[8] Liu J, Zha Z J, Wu W, et al. Spatial-temporal correlation and topology learning for person re-identification in videos[C]//Proceedings of the IEEE/CVF Conference on Computer Vision and Pattern Recognition. 2021: 4370-4379.
We have made changes in response to your comments.The description of this reference is in page 1-line 42-53.
[15] Hu J, Shen L, Sun G. Squeeze-and-excitation networks[C]//Proceedings of the IEEE conference on computer vision and pattern recognition. 2018: 7132-7141.The description of this reference is in page 2-line 68-71.
Point 7:“The cross‐consistency learning module randomly divides the person re‐ identification dataset into two groups based on the camera view” the camera view is not clear to the reader; the authors need to declare the sentences.
Reply:
where [] denotes the downward rounding function. Different camera views can be interpreted specifically as different camera angles. For example, the Market1501 dataset has six cameras, which are randomly divided into two groups of three cameras each, and the DukeMTMC-ReID dataset has eight cameras, which are randomly divided into two groups of four cameras each.
We have made changes in response to your comments. This change is in page 5-line 209-212.
Point 8:What is y1 and y2 in equation 6.
Reply:
where n and m denote the total number of images in each batch of the two sample groups, respectively, p denotes the total number of person categories in the training set, denotes the true label corresponding to the training set image, one denotes the first camera sample set, denotes the true label corresponding to the training set image , two denotes the second camera sample set
We have made changes in response to your comments. This change is in page 6-line 230-232.
Point 9:The symbols inside the text should be in italic format.
Reply:
page 3 - line 118=> " where F'max(x) ∈R1×H×W,F'Avg(x) ∈R1×H×W, " -> " where ∈R1×H×W, ∈R1×H×W,"
We have made changes in response to your comments. This change is in page 4 - line 164.
Reply:
page 4 - line 148-149=> "and Rp denotes a 1×1 convolution operation on the feature map followed by BN normalization and Relu activation." -> "and denotes a 1×1 convolution operation on the feature map followed by BN normalization and Relu activation."
We have made changes in response to your comments. This change is in page 5 - line 191.
Reply: page 5 - line 178=> " at which point P = Q." -> "at which point P = Q."
We have made changes in response to your comments. This change is in page 6 - line 218.
Reply:
page 6- line 186-189=> " where E[w1] and E[w2] denote the " -> " Where E[w1] and E[w2] denote the "
We have made changes in response to your comments. This change is in page 6 - line 249.
Reply:
page 6 - line 188-189=> " are initialized E(0)[w1]=w1 and E(0)[w2]=w2, " -> "are initialized E(0)[w1]=w1 and E(0)[w2]=w2, "
We have made changes in response to your comments. This change is in page 6 - line 251-252.
Reply:
page 6 - line 193-194=> " respectively, and β is the equilibrium parameter of the model update, where β is set to 0.2." -> "respectively, and β is the equilibrium parameter of the model update, where β is set to 0.2."
We have made changes in response to your comments. This change is in page 7 - line 256-257.
Reply:
page 6- line 205-208=> " In this paper, the pedestrian feature maps generated by different layers of the network are denoted by Tk (Tk ∈ T), and the semantic information contained in T1 and T2 is not considered as incomplete. The sizes of the extracted feature maps T3 to T5 have the following sizes,in that order: T3∈R512×32×16, T4∈R1024×16×8 and T5∈R2048×16×8 ." ->
" In this paper, the pedestrian feature maps generated by different layers of the net-work are denoted by (∈ T), and the semantic information contained in T1 and T2 is not considered as incomplete. The sizes of the extracted feature maps T3 to T5 have the following sizes , in that order: T3∈R512×32×16, T4∈R1024×16×8 and T5∈R2048×16×8 "
We have made changes in response to your comments. This change is in page 7 - line 268-271.
Reply:
page 7 - line 235=> "true label yi, W represents the weight" -> "true label , W represents the weight"
We have made changes in response to your comments. This change is in page 8 - line 306.
Reply:
page 7- line 244=> " Where P denotes that there are P images of pedestrians " ->" Where P denotes that there are P images of pedestrians"
We have made changes in response to your comments. This change is in page 8 - line 316.
Reply:
page 7 - line 245=> "K denotes that there are K instances of each identity" -> "K denotes that there are K instances of each identity"
We have made changes in response to your comments. This change is in page 8 - line 317.
Reply:
page 7 - line 247=> " and negative samples, where α = 0.3." -> " and negative samples, where α = 0.3."
We have made changes in response to your comments. This change is in page 8 - line 319.
Point 10:where fa、fp and fn represent the target image, positive sample image and negative sample image respectively, is it positive and negative classified image? The sentences should be clear to understand.
Reply:
Difficult sample triplet loss is a widely used loss function in the field of image identification, where,,denote Anchor pedestrian image features, pedestrian image features with the same label as Anchor, and pedestrian image features with a different label than Anchor, respectively ,
We have made changes in response to your comments. This change is in page 8-line 308-310.

Round 2
Reviewer 2 Report
I suggest accept this paper after finishing several editing errors and double checking English grammar
Author Response
Response to Reviewer 2 Comments
Dear reviewers
Re: Manuscript ID: 1999662 and Title: Joint cross-consistency learning and multi-feature fusion for person re-identification
Thank you for your letter and the reviewers’ comments concerning our manuscript entitled “Joint cross-consistency learning and multi-feature fusion for person re-identification” (1999662). Those comments are valuable and very helpful. We have read through comments carefully and have made corrections. Based on the instructions provided in your letter, we uploaded the file of the revised manuscript. Revisions in the text are shown using red highlight for additions, and strikethrough font for deletions.
We would love to thank you for allowing us to resubmit a revised copy of the manuscript and we highly appreciate your time and consideration.
Sincerely.
Author responses.
Here are our responses and any changes will be indicated in the revised article.
Point 1:I suggest accept this paper after finishing several editing errors and double checking English grammar.
Reply:
page 1 - line 9 => " tasks. this paper proposes a model with a" -> " tasks. This paper proposes a model with a"
We have made changes in response to your comments. This change is in page 1-line 9.
Reply:
page 1 - line 16 => "of data sets, so that the" -> "of datasets, so that the"
We have made changes in response to your comments. This change is in page 1-line 16.
Reply:
page 2 - line 64 => "global features and multi-granularity" -> "global features, and multi-granularity"
We have made changes in response to your comments. This change is in page 2-line 67.
Reply:
page 2 - line 68 => " only attention networks" -> " individual attention networks"
We have made changes in response to your comments. This change is in page 2-line 71.
Reply:
page 2 - line 70-73 => "To address the problems of the above methods, this paper investigates the extraction of pedestrian image features. We designed a model joint cross-consistency learning and multi-feature fusion for person re-identification model CCLM (Cross Consistent Learning and Multi-feature) for person re-identification. The main work of this paper is as follows.
" -> "This study looks into the extraction of pedestrian image features in order to solve the issues with the previous methods. In order to re-identify a person, we created a model called CCLM (Cross Consistent Learning and Multi-Feature), which combines cross-consistency learning and multiple features. The following are the key points of this essay."
We have made changes in response to your comments. This change is in page 2-line 73-76.
Reply:
page 3 - line 102 => " based on the ResNet50 " -> " is based on the ResNet50 "
We have made changes in response to your comments. This change is in page 3-line 106.
Reply:
page 3 - line 103 => "and Multi-feature" -> "and a Multi-feature"
We have made changes in response to your comments. This change is in page 3-line 107.
Reply:
page 3 - line 110 => " procedure divided into" -> " procedure is divided into"
We have made changes in response to your comments. This change is in page 3-line 114.
Reply:
page 3- line 119 => " pyramid module and" -> "pyramid module, and"
We have made changes in response to your comments. This change is in page 3-line 123.
Reply:
page 3 - line 120 => "the triplet loss and" -> " the triplet loss, and"
We have made changes in response to your comments. This change is in page 3-line 124.
Reply:
page 4 - line 152 => " and Conv represents the " -> " and Conv represents the "
We have made changes in response to your comments. This change is in page 4-line 156.
Reply:
page 4 - line 165 => "the ability of person" -> " the ability of the person"
We have made changes in response to your comments. This change is in page 4-line 169.
Reply:
page 4 - line 166 => " information of person" -> " information of the person"
We have made changes in response to your comments. This change is in page 5-line 170.
Reply:
page 5 - line 174 => " with the difference features between" -> " with the different features between"
We have made changes in response to your comments. This change is in page 5-line 178.
Reply:
page 5 - line 183 => " by BN normalization and Relu activation." -> " by BN and Relu activation."
We have made changes in response to your comments. This change is in page 5-line 187.
Reply:
page 5 - line 185-187 => " In order to empower the feature encoder to extract stable features of person images, it is proposed to game different sets of samples crossed with classifiers equipped with specific knowledge." -> "It is suggested to game different sets of samples crossed with classifiers trained with particular knowledge in order to enable the feature encoder to extract reliable characteristics of personal images."
We have made changes in response to your comments. This change is in page 5-line 189-191.
Reply:
page 5- line 195-199 => " As shown in Figure 5, a separate classifier is trained for the feature vectors extracted from person images from different sample groups. Crossing two specific sets of classifiers, given a person image feature vector f, if the two different classifiers output the same prediction, it shows that the network has eliminated the effect of different camera viewpoint transformations." -> " information of the person"
We have made changes in response to your comments. This change is in page 5-line 199-203.
Reply:
page 6 - line 202 => "in the pre-training phase is equation (7)" -> " in the pre-training phase is equation (6)"
We have made changes in response to your comments. This change is in page 6-line 215.
Reply:
page 6 - line 205 => "where n and m denote" -> " Where n and m denote"
We have made changes in response to your comments. This change is in page 6-line 218.
Reply:
page 6 - line 210 => " by classifier w1 and w2 for " -> "by classifier w1 and w2 for "
We have made changes in response to your comments. This change is in page 6-line 223.
Reply:
page 6 - line 218-224 => " The equal sign is obtained when and only when, at which point P = Q. A direct crossover of classifiers with different knowledge would result in a rapid and optimized agreement of the two classifier parameters, so the proposed average camera view classifier with the same structure as the camera view classifier makes it possible to consider not only the parameters of the current batch during the network update, but also the parameters are updated concerning the previous batch, with the crossover consistency loss shown in equation (8)." -> " When and only when , occurs, at which time P = Q, the equal sign is obtained. The proposed average camera view classifier, which has the same structure as the camera view classifier, enables consideration of not only the parameters of the current batch during the network update but also the parameters that are updated concerning the previous batch, with the crossover consistency loss shown in the equation. A direct crossover of classifiers with different knowledge would result in a rapid and optimized agreement of the two classifier parameters (8)."
We have made changes in response to your comments. This change is in page 6-line 231-237.
Reply:
page 7 - line 232 => " Where E(t)[w1] and E(t-1)[w1] denote the model " -> "Where E(t)[w1] and E(t-1)[w1] denote the model "
We have made changes in response to your comments. This change is in page 7-line 278.
Reply:
page 7 - line 237-239 => " During the pre-training phase, the two classifiers' joint cross-entropy loss and cross-consistency learning loss are jointly updated for the backbone network, and the cross-consistency loss balance parameter μ is set to 1.5." -> " During the pre-training phase, the cross-consistency loss balance parameter μ is set to 1.5, and the joint cross-entropy loss and cross-consistency learning loss of the two classifiers are jointly updated for the backbone network."
We have made changes in response to your comments. This change is in page 7-line 283-285.
Reply:
page 7 - line 246 => " contained in T1 and T2 is not" -> "contained in T1 and T2 is not"
We have made changes in response to your comments. This change is in page 7-line 292.
Reply:
page 7 - line 258 => "operation andstands for element-by-element summation" -> " operation ,andstands for element-by-element summation"
We have made changes in response to your comments. This change is in page 7-line 306.
Reply:
page 8 - line 274 => " smoothing shown in equation (15)." -> "smoothing is shown in equation (15)."
We have made changes in response to your comments. This change is in page 8-line 324.
Reply:
page 8 - line 278 => " in each batch, fi represents the" -> "in each batch, fi represents the "
We have made changes in response to your comments. This change is in page 8-line 328.
Reply:
page 8 - line 301 => " information of person" -> " of the same person"
We have made changes in response to your comments. This change is in page 8-line 351.
Reply:
page 9 - line 323 => "Where P denotes the total number of person identities in the same" -> "Where P denotes the total number of personal identities in the same"
We have made changes in response to your comments. This change is in page 9-line 374.
Reply:
page 12 - line 438 => " images of person unaligned" -> " images of the person unaligned"
We have made changes in response to your comments. This change is in page 12-line 489.
Reply:
page 12 - line 438 => " person occluded" -> "person occluded, "
We have made changes in response to your comments. This change is in page 12-line 489.
Reply:
page 12 - line 442 => "and other person information, " -> " and other personal information, "
We have made changes in response to your comments. This change is in page 12-line 493.
Reply:
page 14 - line 487 => "pre-training, due to the fact that the influence of " -> "pre-training, since the influence of "
We have made changes in response to your comments. This change is in page 14-line 538.
Reply:
Page 14 - line 505 => "Center denotes the cluster loss. " -> " Center denotes the cluster center loss. "
We have made changes in response to your comments. This change is in page 14-line 556.
Reply:
page 15 - line 523 => " The hyperparameters γ and m of Cosine Softmax Loss and the hyperparameter ρ of cluster center" -> " The parameters γ and m of Cosine Softmax Loss and the parameter ρ of cluster center"
We have made changes in response to your comments. This change is in page 15-line 575.
Reply:
page 15 - line 526 => "and function, rank-1 and " -> " function, rank-1 and "
We have made changes in response to your comments. This change is in page 15-line 578.
Reply:
page 15 - line 530 => " metric is higher for ρ values " -> " metric is higher for ρ values "
We have made changes in response to your comments. This change is in page 15-line 582.
Reply:
page 15 - line 548 => " multi-feature fusion divide into " -> " multi-feature fusion is divided into "
We have made changes in response to your comments. This change is in page 15-line 600.
Reply:
page 16 - line 557=> "and lightweighting models in conjunction " -> "and lightweight models in conjunction "
We have made changes in response to your comments. This change is in page 16-line 613.

Reviewer 3 Report
however the paper has been improved in the revised version, there are many grammatical, typo, and Capital letters inside the text. I would suggested the revise the final version carefully
Author Response
Response to Reviewer 3 Comments
Dear reviewers
Re: Manuscript ID: 1999662 and Title: Joint cross-consistency learning and multi-feature fusion for person re-identification
Thank you for your letter and the reviewers’ comments concerning our manuscript entitled “Joint cross-consistency learning and multi-feature fusion for person re-identification” (1999662). Those comments are valuable and very helpful. We have read through comments carefully and have made corrections. Based on the instructions provided in your letter, we uploaded the file of the revised manuscript. Revisions in the text are shown using red highlight for additions, and strikethrough font for deletions.
We would love to thank you for allowing us to resubmit a revised copy of the manuscript and we highly appreciate your time and consideration.
Sincerely.
Author responses.
Here are our responses and any changes will be indicated in the revised article.
Point 1:however the paper has been improved in the revised version, there are many grammatical, typo, and Capital letters inside the text. I would suggested the revise the final version carefully.
Reply:
page 1 - line 9 => " tasks. this paper proposes a model with a" -> " tasks. This paper proposes a model with a"
We have made changes in response to your comments. This change is in page 1-line 9.
Reply:
page 1 - line 16 => "of data sets, so that the" -> "of datasets, so that the"
We have made changes in response to your comments. This change is in page 1-line 16.
Reply:
page 2 - line 64 => "global features and multi-granularity" -> "global features, and multi-granularity"
We have made changes in response to your comments. This change is in page 2-line 67.
Reply:
page 2 - line 68 => " only attention networks" -> " individual attention networks"
We have made changes in response to your comments. This change is in page 2-line 71.
Reply:
page 2 - line 70-73 => "To address the problems of the above methods, this paper investigates the extraction of pedestrian image features. We designed a model joint cross-consistency learning and multi-feature fusion for person re-identification model CCLM (Cross Consistent Learning and Multi-feature) for person re-identification. The main work of this paper is as follows.
" -> "This study looks into the extraction of pedestrian image features in order to solve the issues with the previous methods. In order to re-identify a person, we created a model called CCLM (Cross Consistent Learning and Multi-Feature), which combines cross-consistency learning and multiple features. The following are the key points of this essay."
We have made changes in response to your comments. This change is in page 2-line 73-76.
Reply:
page 3 - line 102 => " based on the ResNet50 " -> " is based on the ResNet50 "
We have made changes in response to your comments. This change is in page 3-line 106.
Reply:
page 3 - line 103 => "and Multi-feature" -> "and a Multi-feature"
We have made changes in response to your comments. This change is in page 3-line 107.
Reply:
page 3 - line 110 => " procedure divided into" -> " procedure is divided into"
We have made changes in response to your comments. This change is in page 3-line 114.
Reply:
page 3- line 119 => " pyramid module and" -> "pyramid module, and"
We have made changes in response to your comments. This change is in page 3-line 123.
Reply:
page 3 - line 120 => "the triplet loss and" -> " the triplet loss, and"
We have made changes in response to your comments. This change is in page 3-line 124.
Reply:
page 4 - line 152 => " and Conv represents the " -> " and Conv represents the "
We have made changes in response to your comments. This change is in page 4-line 156.
Reply:
page 4 - line 165 => "the ability of person" -> " the ability of the person"
We have made changes in response to your comments. This change is in page 4-line 169.
Reply:
page 4 - line 166 => " information of person" -> " information of the person"
We have made changes in response to your comments. This change is in page 5-line 170.
Reply:
page 5 - line 174 => " with the difference features between" -> " with the different features between"
We have made changes in response to your comments. This change is in page 5-line 178.
Reply:
page 5 - line 183 => " by BN normalization and Relu activation." -> " by BN and Relu activation."
We have made changes in response to your comments. This change is in page 5-line 187.
Reply:
page 5 - line 185-187 => " In order to empower the feature encoder to extract stable features of person images, it is proposed to game different sets of samples crossed with classifiers equipped with specific knowledge." -> "It is suggested to game different sets of samples crossed with classifiers trained with particular knowledge in order to enable the feature encoder to extract reliable characteristics of personal images."
We have made changes in response to your comments. This change is in page 5-line 189-191.
Reply:
page 5- line 195-199 => " As shown in Figure 5, a separate classifier is trained for the feature vectors extracted from person images from different sample groups. Crossing two specific sets of classifiers, given a person image feature vector f, if the two different classifiers output the same prediction, it shows that the network has eliminated the effect of different camera viewpoint transformations." -> " information of the person"
We have made changes in response to your comments. This change is in page 5-line 199-203.
Reply:
page 6 - line 202 => "in the pre-training phase is equation (7)" -> " in the pre-training phase is equation (6)"
We have made changes in response to your comments. This change is in page 6-line 215.
Reply:
page 6 - line 205 => "where n and m denote" -> " Where n and m denote"
We have made changes in response to your comments. This change is in page 6-line 218.
Reply:
page 6 - line 210 => " by classifier w1 and w2 for " -> "by classifier w1 and w2 for "
We have made changes in response to your comments. This change is in page 6-line 223.
Reply:
page 6 - line 218-224 => " The equal sign is obtained when and only when, at which point P = Q. A direct crossover of classifiers with different knowledge would result in a rapid and optimized agreement of the two classifier parameters, so the proposed average camera view classifier with the same structure as the camera view classifier makes it possible to consider not only the parameters of the current batch during the network update, but also the parameters are updated concerning the previous batch, with the crossover consistency loss shown in equation (8)." -> " When and only when , occurs, at which time P = Q, the equal sign is obtained. The proposed average camera view classifier, which has the same structure as the camera view classifier, enables consideration of not only the parameters of the current batch during the network update but also the parameters that are updated concerning the previous batch, with the crossover consistency loss shown in the equation. A direct crossover of classifiers with different knowledge would result in a rapid and optimized agreement of the two classifier parameters (8)."
We have made changes in response to your comments. This change is in page 6-line 231-237.
Reply:
page 7 - line 232 => " Where E(t)[w1] and E(t-1)[w1] denote the model " -> "Where E(t)[w1] and E(t-1)[w1] denote the model "
We have made changes in response to your comments. This change is in page 7-line 278.
Reply:
page 7 - line 237-239 => " During the pre-training phase, the two classifiers' joint cross-entropy loss and cross-consistency learning loss are jointly updated for the backbone network, and the cross-consistency loss balance parameter μ is set to 1.5." -> " During the pre-training phase, the cross-consistency loss balance parameter μ is set to 1.5, and the joint cross-entropy loss and cross-consistency learning loss of the two classifiers are jointly updated for the backbone network."
We have made changes in response to your comments. This change is in page 7-line 283-285.
Reply:
page 7 - line 246 => " contained in T1 and T2 is not" -> "contained in T1 and T2 is not"
We have made changes in response to your comments. This change is in page 7-line 292.
Reply:
page 7 - line 258 => "operation andstands for element-by-element summation" -> " operation ,andstands for element-by-element summation"
We have made changes in response to your comments. This change is in page 7-line 306.
Reply:
page 8 - line 274 => " smoothing shown in equation (15)." -> "smoothing is shown in equation (15)."
We have made changes in response to your comments. This change is in page 8-line 324.
Reply:
page 8 - line 278 => " in each batch, fi represents the" -> "in each batch, fi represents the "
We have made changes in response to your comments. This change is in page 8-line 328.
Reply:
page 8 - line 301 => " information of person" -> " of the same person"
We have made changes in response to your comments. This change is in page 8-line 351.
Reply:
page 9 - line 323 => "Where P denotes the total number of person identities in the same" -> "Where P denotes the total number of personal identities in the same"
We have made changes in response to your comments. This change is in page 9-line 374.
Reply:
page 12 - line 438 => " images of person unaligned" -> " images of the person unaligned"
We have made changes in response to your comments. This change is in page 12-line 489.
Reply:
page 12 - line 438 => " person occluded" -> "person occluded, "
We have made changes in response to your comments. This change is in page 12-line 489.
Reply:
page 12 - line 442 => "and other person information, " -> " and other personal information, "
We have made changes in response to your comments. This change is in page 12-line 493.
Reply:
page 14 - line 487 => "pre-training, due to the fact that the influence of " -> "pre-training, since the influence of "
We have made changes in response to your comments. This change is in page 14-line 538.
Reply:
Page 14 - line 505 => "Center denotes the cluster loss. " -> " Center denotes the cluster center loss. "
We have made changes in response to your comments. This change is in page 14-line 556.
Reply:
page 15 - line 523 => " The hyperparameters γ and m of Cosine Softmax Loss and the hyperparameter ρ of cluster center" -> " The parameters γ and m of Cosine Softmax Loss and the parameter ρ of cluster center"
We have made changes in response to your comments. This change is in page 15-line 575.
Reply:
page 15 - line 526 => "and function, rank-1 and " -> " function, rank-1 and "
We have made changes in response to your comments. This change is in page 15-line 578.
Reply:
page 15 - line 530 => " metric is higher for ρ values " -> " metric is higher for ρ values "
We have made changes in response to your comments. This change is in page 15-line 582.
Reply:
page 15 - line 548 => " multi-feature fusion divide into " -> " multi-feature fusion is divided into "
We have made changes in response to your comments. This change is in page 15-line 600.
Reply:
page 16 - line 557=> "and lightweighting models in conjunction " -> "and lightweight models in conjunction "
We have made changes in response to your comments. This change is in page 16-line 613.
